# Unraveling the Antioxidant Activity of *2R*,*3R*-dihydroquercetin

**DOI:** 10.3390/ijms241814220

**Published:** 2023-09-18

**Authors:** Yaping Xu, Zhengwen Li, Yue Wang, Chujie Li, Ming Zhang, Haiming Chen, Wenxue Chen, Qiuping Zhong, Jianfei Pei, Weijun Chen, Guido R. M. M. Haenen, Mohamed Moalin

**Affiliations:** 1College of Food Science and Engineering, Hainan University, 58 Renmin Road, Haikou 570228, China; xyp18161011146@163.com (Y.X.); 992984@hainanu.edu.cn (H.C.); hnchwx@vip.163.com (W.C.); hainufood88@163.com (Q.Z.); peijianfei@hainanu.edu.cn (J.P.); 2School of Pharmacy, Chengdu University, 2025 Chengluo Avenue, Chengdu 610106, China; lizhengwen@cdu.edu.cn; 3Department of Pharmacology and Personalized Medicine, School of Nutrition and Translational Research in Metabolism (NUTRIM), Cardiovascular Research Institute Maastricht (CARIM), Faculty of Health, Medicine and Life Sciences, Maastricht University, 6200 MD Maastricht, The Netherlands; yue.wang@maastrichtuniversity.nl (Y.W.); c.li@maastrichtuniversity.nl (C.L.); g.haenen@maastrichtuniversity.nl (G.R.M.M.H.); 4Research Centre Material Sciences, Zuyd University of Applied Science, 6400 AN Heerlen, The Netherlands; maxamed.moalin@gmail.com

**Keywords:** *2R*,*3R*-dihydroquercetin, redox modulation, quercetin, quinone, epimerization

## Abstract

It has been reported that in an oxidative environment, the flavonoid *2R*,*3R*-dihydroquercetin (*2R*,*3R*-DHQ) oxidizes into a product that rearranges to form quercetin. As quercetin is a very potent antioxidant, much better than *2R*,*3R*-DHQ, this would be an intriguing form of targeting the antioxidant quercetin. The aim of the present study is to further elaborate on this targeting. We can confirm the previous observation that *2R*,*3R*-DHQ is oxidized by horseradish peroxidase (HRP), with H_2_O_2_ as the oxidant. However, HPLC analysis revealed that no quercetin was formed, but instead an unstable oxidation product. The inclusion of glutathione (GSH) during the oxidation process resulted in the formation of a *2R*,*3R*-DHQ-GSH adduct, as was identified using HPLC with IT-TOF/MS detection. GSH adducts appeared on the B-ring of the *2R*,*3R*-DHQ quinone, indicating that during oxidation, the B-ring is oxidized from a catechol to form a quinone group. Ascorbate could reduce the quinone back to *2R*,*3R*-DHQ. No *2S*,*3R*-DHQ was detected after the reduction by ascorbate, indicating that a possible epimerization of *2R*,*3R*-DHQ quinone to *2S*,*3R*-DHQ quinone does not occur. The fact that no epimerization of the oxidized product of *2R*,*3R*-DHQ is observed, and that GSH adducts the oxidized product of *2R*,*3R*-DHQ on the B-ring, led us to conclude that the redox-modulating activity of *2R*,*3R*-DHQ quinone resides in its B-ring. This could be confirmed by chemical calculation. Apparently, the administration of *2R*,*3R*-DHQ in an oxidative environment does not result in ‘biotargeting’ quercetin.

## 1. Introduction

Redox energy is essential for life, as it fuels all our biochemical processes [1]. Most redox energy is safely channeled in the biochemical networks, but for some unknown reason, part of the oxidizing energy flow is uncontrolled [2]. This disordered energy can, seemingly quite randomly, damage our cells, and result in oxidative-stress-related diseases such as cardiovascular disease, arthritis, cancer, atherosclerosis, aging, and neurological and neurodegenerative disorders [3,4]. In a healthy body, the oxidizing and reducing forces are—to a large extent—balanced and generate a reasonably controlled energy flow in which the accumulation of oxidative damage is kept within limits, and we can enjoy a lifespan of, on average, 80 years [5]. Intriguingly, the disordered energy also has a ‘good’ side, as it is involved in cell signaling, e.g., it can lead to adaptation of cells, making them more resilient [6]. 

Antioxidants might be used to modulate and redirect the disordered redox energy [7]. To do this, the antioxidants have to be present at the right place and at the right moment, e.g., within the proper time window at locations where the energy is disordered. An example of this targeting of antioxidants is that quercetin can be set free from quercetin-glucuronides at the site of inflammation by glucuronidases that are released by inflammatory cells [8,9]. Quercetin has a higher redox-modulating potency than its glucuronide, and thus ‘liberating’ quercetin will result in bioactiviation [10]. Moreover, quercetin is more lipophilic compared with its glucuronide, and will not be that effectively removed by the blood at the site of inflammation [11]. Both the bioactivation as well as the retention of quercetin result in targeting of the redox modulator at the site of inflammation, the location where it then can display its anti-inflammatory activity [12].

Another targeting strategy that has been proposed is that the oxidative environment produced by the disordered energy may lead to the bioactivation of antioxidants [13]. Rogozhin et al. have reported that by scavenging oxidative species, the flavonoid *2R*,*3R*-dihydroquercetin (*2R*,*3R*-DHQ) is oxidized to form a quinone-like product that rearranges to create quercetin (Figure 1) [14]. *2R*,*3R*-DHQ lacks the C2–C3 double bound, which is one of the main requirements for a flavonoid to act as a good antioxidant [15,16]. Quercetin contains this C2–C3 double bound. In most assays, quercetin is the most potent antioxidant [17]. This has been linked to its anti-inflammatory, anti-diabetic, anti-fibrotic, anti-coagulative, anti-bacterial, anti-atherogenic, anti-hypertensive, and anti-proliferative activities observed in numerous in vivo and in vitro studies [18,19]. This would mean that the conversion of *2R*,*3R*-DHQ to quercetin results in the targeting of the redox modulator in an oxidative environment, the location where quercetin can then display its potent redox-modulating activity. The aim of the present study is to evaluate this hypothesis and to further elaborate the redox-modulating activity of *2R*,*3R*-DHQ.

## 2. Results

### 2.1. Oxidation of 2R,3R-DHQ by H_2_O_2_ and HRP Does Not Results in the Formation of Quercetin 

Addition of H_2_O_2_ and HRP to the *2R*,*3R*-DHQ solution resulted in the oxidation of *2R*,*3R*-DHQ (Figure 2A). The decrease in UV spectrum at 330 nm (the λmax of *2R*,*3R*-DHQ) demonstrated that *2R*,*3R*-DHQ was consumed, which is consistent with the decrease in the *2R*,*3R*-DHQ peak at 12.7 min in the HPLC chromatogram of the incubation mixture (Figure 3B). Moreover, the reaction of *2R*,*3R*-DHQ with H_2_O_2_ and HRP leads to an increase in absorbance at 420 nm and 270 nm, which can be ascribed to the formation of oxidation product(s). The results for both HPLC and LCMS-IT-TOF showed no detectible quercetin formation in the incubation mixture. Based on analogy with other studies on, e.g., the oxidation of quercetin and epicatechin, the oxidation product of *2R*,*3R*-DHQ might be *2R*,*3R*-DHQ quinone [20,21]. However, the *2R*,*3R*-DHQ quinone was not successfully detected either by HPLC or LCMS-IT-TOF/MS. This indicates the instability of *2R*,*3R*-DHQ quinone, which is comparable to the instability of quercetin quinone and (-)-epicatechin quinone [22]. The formation of the latter quinones was proven by trapping the quinone with GSH; therefore, we here used the same strategy.

### 2.2. Trapping the Quinone Formed in the Oxidation of 2R,3R-DHQ by GSH

When *2R*,*3R*-DHQ was oxidized by H_2_O_2_ and HRP in the presence of GSH, an isosbestic point was seen in the UV spectrum at 280 nm, which was not present in the UV spectrum of the incubation mixture without GSH. This indicates that a GSH adduct is formed (Figure 2C). HPLC analysis of the incubation mixture with GSH showed a peak in the chromatogram at 10.7 min (Figure 3D), which is not found in the HPLC chromatogram of the incubation mixture without GSH. IT-TOF/MS analysis of the incubation of *2R*,*3R*-DHQ with H_2_O_2_, HRP and GSH confirmed the presence of *2R*,*3R*-DHQ-GSH adduct. The protonated and fragmented ions of *2R*,*3R*-DHQ-GSH adduct were 272.0853 *m*/*z*, 317.0127 *m*/*z* and 335.0218 *m*/*z*, whereas *2R*,*3R*-DHQ yielded fragment ions at 125.0284 *m*/*z*, 175.0424 *m*/*z*, 177.0221 *m*/*z*, 241.0525 *m*/*z* and 285.0430 *m*/*z* (Figure 4). The formation of *2R*,*3R*-DHQ-GSH adduct demonstrates that H_2_O_2_ and HRP converts *2R*,*3R*-DHQ into its quinone. 

### 2.3. Ascorbate Regenerates 2R,3R-DHQ Quinone to 2R,3R-DHQ 

The addition of ascorbate to the incubation mixture containing *2R*,*3R*-DHQ, H_2_O_2_, and HRP did not result in a net consumption of *2R*,*3R*-DHQ, since the absorbance of UV spectrum between 300 and 400 nm remained unchanged (Figure 2B). However, ascorbate was consumed, which was evidenced by the decrease in absorbance at 265 nm (λmax of ascorbate). The parallel control experiment showed that there was much less ascorbate consumption in the incubation of ascorbate, H_2_O_2_ and HRP (Figure 5A). Therefore, it is concluded that *2R*,*3R*-DHQ is oxidized by H_2_O_2_ and HRP into *2R*,*3R*-DHQ quinone, and the quinone is then regenerated by ascorbate into *2R*,*3R*-DHQ before the quinone is degraded. Assuming that the reaction of ascorbate and *2R*,*3R*-DHQ quinone is 1:1, the rate of *2R*,*3R*-DHQ quinone formation is equal to ascorbate consumption by H_2_O_2_ and HRP, which is 1.61 ± 0.10 µM/min (Figure 5A). Moreover, HPLC analysis confirmed that there was no net *2R*,*3R*-DHQ consumption in the incubation system when ascorbate was present in the incubation (0.04 ± 0.02 μM) (Figure 3C). This rate of *2R*,*3R*-DHQ quinone formation is similar to the rate of *2R*,*3R*-DHQ consumption in the incubation system in which *2R*,*3R*-DHQ was oxidized by H_2_O_2_ and HRP without ascorbate, namely 14.35 ± 0.26 μM in 8.5 min, which is 1.68 ± 0.03 μM/min (Figure 3B and Figure 4C). 

So, when *2R*,*3R*-dihdroquercetin is oxidized by H_2_O_2_ and HRP in presence of ascorbate, *2R*,*3R*-DHQ quinone is formed. The quinone might epimerize as shown in Figure 6. In the quinone intermediate, the C2-C1′ bound changes from a single bound to a double bound. Converting this double bound back to a single bound can be achieved in two manners and therefore leads to (partial) epimerization at the asymmetric C2 carbon. This would result in (partial) conversion of *2R*,*3R*-DHQ quinone to *2R*,*3S*-DHQ quinone. Due to the limited stability of the quinone itself, this could not be investigated directly by examining the quinone itself.

As shown above, in presence of ascorbate, the quinone is reduced before it is degraded. If the epimerization of the quinone would take place and thus *2R*,*3R*-DHQ quinone and *2S*,*3R*-DHQ quinone are formed, the reduction of *2S*,*3R*-DHQ quinone by ascorbate will yield *2S*,*3R*-DHQ, and the reduction of *2R*,*3R*-DHQ quinone by ascorbate will yield *2R*,*3R*-DHQ (Figure 6). *2R*,*3R*-DHQ and 2S,3R-DHQ can be identified separately using HPLC since they are diastereoisomers (Appendix A) [23]. However, in the incubation mixture containing *2R*,*3R*-DHQ, H_2_O_2_, HRP and ascorbate, only the *2R*,*3R* isomer of DHQ was detected by HPLC, with no other appreciable peaks. Apparently, there is no substantial epimerization of *2R*,*3R*-DHQ quinone to *2S*,*3R*-DHQ quinone. 

### 2.4. The HOMO/LUMO Energy and the Dual Descriptor of DHQ Quinone

Computer theoretical calculations were performed to confirm the results obtained. The HOMO-LUMO gap energy and the single point energy can show the energy barrier between compounds, and prove whether the isomerization of DHQ quinone exists or not. For the epimerization of *2R*,*3R*-DHQ quinone, the SP3 nature of the hybridisation at the C2 carbon has to be converted into SP2, which most probably would involve the creation of a double bound between the C2 carbon and the C1′ carbon on B-ring (the conversion of the *2R*,*3R*-DHQ quinone into a flat quinone intermediate). However, during optimization of the flat quinone intermediate in the calculation system, the C2 carbon mostly prefers to form a single bond with C1′ carbon. This indicates that the formation of a double bond between the C2 carbon and the C1′ carbon is not likely. Therefore, the epimerization on the C2 carbon does not occur, and the conversion of the *2R*,*3R*-DHQ quinone into a flat quinone intermediate is not energically preferable. This corroborates the experimental findings that there is no isomerization of *2R*,*3R*-DHQ quinone to *2S*,*3R*-DHQ quinone, and no epimerization of *2R*,*3R*-DHQ to *2S*,*3R*-DHQ when the quinone was reduced by ascorbate. In the literature [24,25,26], the open quinone was proposed as intermediate in the epimerization of *2R*,*3R*-DHQ to *2S*,*3R*-DHQ. Therefore, the open quinone is also considered as intermediate in the oxidation and reduction of *2R*,*3R*-DHQ. It was found that the energy barrier between *2R*,*3R*-DHQ quinone and the open quinone was very high, similar to the energy barrier between the open quinone and *2S*,*3R*-DHQ quinone (Appendix A). Again, this corroborates that in our experiment, the epimerization of *2R*,*3R*-DHQ quinone does not occur. 

The dual descriptor was utilized to locate the redox-active sites on *2R*,*3R*-DHQ quinone, where the nucleophilic and electrophilic reactions are prone to happen. In the dual descriptor of *2R*,3R-DHQ quinone, the green color represents the nucleophilic reaction site, and the blue color represents the electrophilic reaction site. GSH is a nucleophilic compound and therefore will adduct to the nucleophilic reaction site *2R*,*3R*-DHQ quinone. In the dual descriptor of *2R*,*3R*-DHQ quinone, the C2′ and C5′ position in the B-ring were the greenest, indicating that GSH will mostly adduct there (Appendix A). This further corroborates that the chemical reactivity of *2R*,*3R*-DHQ quinone resides in its B-ring. 

## 3. Discussion

To confirm the reported finding that *2R*,*3R*-DHQ is converted to quercetin in an oxidative environment, we repeated the experiment of Rogozhin et al., who oxidized *2R*,*3R*-DHQ with HRP/ H_2_O_2_ and reported the formation of quercetin [14]. We could reproduce the conversion of *2R*,*3R*-DHQ is into a reaction product by HRP/H_2_O_2_. Rogozhin et al. analyzed the incubation mixture by recording the change in absorption and concluded that the product was quercetin based on the absorption maximum at 420–440 nm [14]. We also found that the highest rise in absorption in the reaction mixture was between 420 and 440 nm. However, HPLC analysis showed that no quercetin was formed. Based on the HRP/H_2_O_2_ oxidation of similar flavonoids, i.e., quercetin and (-)-epicatechin, we assumed that *2R*,*3R*-DHQ is converted to *2R*,*3R*-DHQ quinone. The quinones of quercetin and (-)-epicatechin could not be detected using HPLC, due to a limited stability, and the formation of these quinones was demonstrated by trapping the quinones with GSH and detecting the stable GSH adducts [20,27]. Using the same strategy, we were able to detect a *2R*,*3R*-DHQ-GSH adduct, demonstrating that in the oxidation of *2R*,*3R*-DHQ also a quinone is formed. 

We also studied the possible epimerization of *2R*,*3R*-DHQ quinone to *2S*,*3R*-DHQ quinone. The limited stability of the quinone prevented direct analysis of the quinone [28], and therefore we could not investigate this by looking at the epimerization of the quinone itself. We therefore investigated the reaction products of the ‘in situ’ generated quinone with GSH and ascorbate. We found that after oxidation and trapping with GSH, only one GSH adduct was found using HLPC analysis. After oxidation and reduction of the quinone by ascorbate, only *2R*,*3R*-DHQ was detected. When epimerization of the *2R*,*3R*-DHQ-quinone would have occurred, two GHS-adduct should have been formed, and after reduction by ascorbate beside *2R*,*3R*-DHQ also *2R*,*3S*-DHQ should have been formed. Since neither occurred, we conclude that there is no appreciable amount of epimerization of *2R*,*3R*-DHQ quinone to *2R*,*3S*-DHQ quinone under the conditions we used. We could corroborate this with chemical calculation, that shows that the energy barrier of the epimerization is too high.

The findings that there is no appreciable amount of epimerization and GSH adducts of the *2R*,*3R*-DHQ-quonone on its B-ring, indicate that the oxidative energy is mainly retained in B-ring. In respect to all the reactions described in this manuscript, *2R*,*3R*-DHQ behaves similar to (-)-epicatechin [20]. (-)-Epicatechin is a flavanol that, compared with the dihydroflavonol *2R*,*3R*-DHQ, lacks the double-bound oxygen on C4 [29]. Apparently, the double bound oxygen, which is located on the AC-ring, has no drastic influence. This also indicates that the B-ring is the pivotal moiety in the redox activity of *2R*,*3R*-DHQ, as well as (-)-epicatechin. 

As also indicated above, the only difference between *2R*,*3R*-DHQ and quercetin is the presence of a double bound between C2 and C3. Due to the presence of this double bound, quercetin has a large conjugated system over the entire backbone of the molecule [30], whereas in *2R*,*3R*-DHQ, the conjugated systems of the AC-ring and the B-ring are not connected. The large system is one of the characteristics that makes quercetin a very efficient antioxidant. Moreover, due to this large system, the oxidative energy can flow over the whole molecule [31]. The flexibility of the flow of the energy through the entire quercetin molecule adds to its efficiency and increases the versatility of quercetin’s redox-modulating potency [32,33]. For example, GSH will adduct quercetin quinone not only at one position, but at both position 6 and 8 of the A-ring [34]. With the electron density concentrated in its B-ring, *2R*,*3R*-DHQ quinone is harder than quercetin quinone. 

The chemical reactivity of the oxidized antioxidant is important, because oxidized antioxidants that chemically seen is an electrophile can pass the disordered energy on to other antioxidants such as GSH and ascorbate, or to redox switches [35]. Directing the disordered energy by the redox-modulating antioxidant to turn on redox switches will result in a major adaptive response of cells. Therefore, not only the difference in chemical reactivity between *2R*,*3R*-DHQ, quercetin and other flavonoids should be considered, also how these compounds are ‘bioactivated’ to their corresponding electrophilic quinones in an oxidative environment and differently redirect the disordered energy should be considered in ‘understanding’ their biological effects. According to Pearson’s Hard and Soft Acids and Bases concept: “hard” electrophiles will more likely react with “hard” nucleophiles, and “soft” electrophiles will more likely react with “soft” nucleophiles [36]. However, this is still mainly “terra incognita”. Identifying the electrophilic oxidation products of specific flavonoids is a step further in the differentiation between bioactive flavonoids to finally come up with a rational for selecting the appropriate flavonoid for a specific disorder. 

## 4. Materials and Methods

### 4.1. Chemicals

*2R*,*3R*-DHQ, hydrogen peroxide (H_2_O_2_), horseradish peroxidase (HRP), ascorbate, glutathione (GSH), and quercetin were purchased from Aladdin Biochemical Technology Co., Ltd. (Shanghai, China), and dissolved in assay buffer (100 mM ammonium bicarbonate, pH 7.4). 

### 4.2. 2R,3R-DHQ Oxidation

A *2R*,*3R*-DHQ stock solution (1 mM) in ethanol (99% *v*/*v*) was prepared and diluted in assay buffer. The oxidation of *2R*,*3R*-DHQ was performed at 37 ℃ in an assay buffer that contained 50 µM *2R*,*3R*-DHQ, 3.2 nM HRP and 33 µM H_2_O_2_. In parallel, the same experiments were performed in the presence of 50 µM ascorbate or 100 µM GSH. After starting the reaction by adding HRP, 5 UV absorption spectrum (λ = 200–600 nm) was recorded with 2 min intervals [37]. Ascorbate consumption was determined from the reduction in the absorption at 265 nm using a molar extinction coefficient of 12,840 M^−1^·cm^−1^. As control, UV absorption of 5 UV scans of the above-mentioned single compounds were recorded. The results showed no change of the absorption spectrum in time and are therefore not shown. Additionally, reaction mixtures were analyzed 8.5 min after starting the reaction with high-performance liquid chromatography (HPLC) and LCMS-ESI-IT-TOF/MS. During the 8.5 min, the incubation mixture was spectrophotometrically monitored to check the progress of the reaction. 

### 4.3. HPLC Analysis

HPLC of the incubation mixtures was performed using Agilent HPLC 1260 system and a diode array detector (Agilent Technologies, Santa Clara, CA, USA). An ZORBAX SB-C18 column (250 × 4.6 mm, 5 µm) was used. The column was eluted with a mixture of distilled water containing 1% (*v*/*v*) acetic acid (solvent A) and methanol (solvent B). The gradient program was: 0–7 min: 5–40% B; 7–27 min: 40–55% B. After each injection, 55% methanol was used to wash the column for 5 min, and the column was reequilibrated with 5% methanol for 5 min. The flow rate was 1.0 mL/min, and 10 µL of samples were injected. The chromatograms presented were based on the detection at 270 nm [38].

### 4.4. LCMS-ESI-IT-TOF/MS and MS/MS Analysis

The reaction products were separated using a Shimadzu HPLC system (Shimadzu technologies, Japan), consisting of PDA diode array detector and Ion-trap time-of-flight tandem mass spectrometry (IT-TOF). An ACQUITY UPLC^®^BEH C18 (2.1 × 100 mm, 1.7 µm) column was selected as the stationary phase. The mobile phase consists of a mixture of 0.1% (*v*/*v*) formic acid in water (solvent A) and methanol (solvent B). The gradient elution started with 10% B and increased linearly to 95% B over 10.1 min before decreasing to 10% B over 5 min. The flow rate was set to 0.35 mL/min, and the column temperature was maintained at 37 °C. An equilibrium period of 3.9 min was used between the two injections. The injection volume was 10 µL [39].

The mass spectrometry experiments were performed with an electrospray ionization (ESI) source operated in both positive and negative ion modes. The IT-TOF/MS was operated using ESI as follows: positive ion voltage of 4.5 kV, negative ion voltage of −3.5 kV, and detector voltage of 1.6 kV. Nitrogen was used as the nebulizing dry gas, and argon was used as the collision gas. The nebulizing gas flow was 1.5 L/min, and the dry gas pressure was 114 kPa. Both the curved desolvation line (CDL) temperature and the block heater temperature were 200 °C. The *m*/*z* scan ranged from 100 to 1000 [40,41].

The eluate peaks corresponding to *2R*,*3R*-DHQ and *2R*,*3R*-DHQ-GSH adduct were analyzed via IT-TOF MS/MS. In negative ion mode, the scanning range of secondary mass spectrometry was 50–320 *m*/*z* and 250–700 *m*/*z*. The parent ions were crushed at 303.0530 *m*/*z* and 608.1192 *m*/*z* to generate characteristic ions. The ion accumulation time was 10 ms. Both the CID collision energy and collision gas energy were 50%. The mass number deviation was lower than 15 ppm.

### 4.5. The HOMO/LUMO Energy and the Dual Descriptor of DHQ Quinone

The equilibrium geometries of all compounds were optimized by Gaussian 09 package [42] using the DFT method at the M062X [43]/6-311+G (d,p) [44] level, whereas Grimme’s DFT-D3 dispersion correction was also employed [45]. The solvent effects on the tested compounds were taken into account via the application of the implemented Solvation Model Density (SMD, Water) method [46]. The dual descriptor was generated by the methods implemented in Multiwfn [47]. Graphical pictures of the dual descriptor, Highest Occupied Molecular Orbital (HOMO) map, and Lowest Unoccupied Molecular Orbital (LUMO) map were generated with the help of Multiwfn [48] and VMD [49]. 

### 4.6. Statistics

All experiments were performed at least in triplicate. Data were given as mean ± standard error of the mean (S.E.M.), or as a typical example. 

## 5. Conclusions

During oxidation of *2R*,*3R*-DHQ, its B-ring is oxidized and *2R*,*3R*-DHQ quinone is formed. This will not lead to the formation of quercetin; thus, ‘biotargeting’ of quercetin from *2R*,*3R*-DHQ in an oxidative environment does not occur. There is no isomerization of *2R*,*3R*-DHQ quinone to *2S*,*3R*-DHQ quinone. It appears that the redox activity of *2R*,*3R*-DHQ quinone resides in its B-ring. 

## Figures and Tables

**Figure 1 ijms-24-14220-f001:**
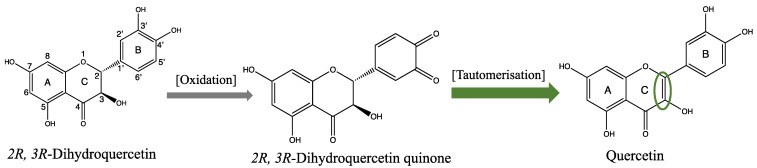
Proposed ‘bioactivation’ of *2R*,*3R*-dihydroquercetin (*2R*,*3R*-DHQ) in an oxidative environment to quercetin. Quercetin possesses a C2–C3 double bound (indicated by the green ellipse), which makes it a better antioxidant than *2R*,*3R*-DHQ.

**Figure 2 ijms-24-14220-f002:**
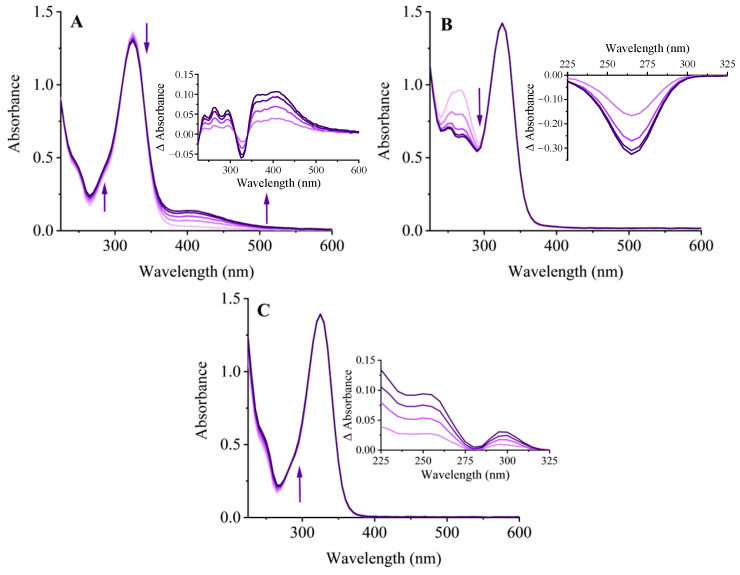
UV scans of an incubation mixture containing 50 µM *2R*,*3R*-DHQ, 3.2 nM HRP and 33 µM H_2_O_2_ (**A**); the incubation mixture of A also containing 50 µM ascorbate (**B**); the incubation mixture of A also containing 100 µM GSH (**C**). The reaction was started by the addition of HRP. Directly after starting the reaction, five UV scans were made at 2 min intervals. The insert shows the difference in absorption of each scan compared with that of the first scan. A typical example is presented.

**Figure 3 ijms-24-14220-f003:**
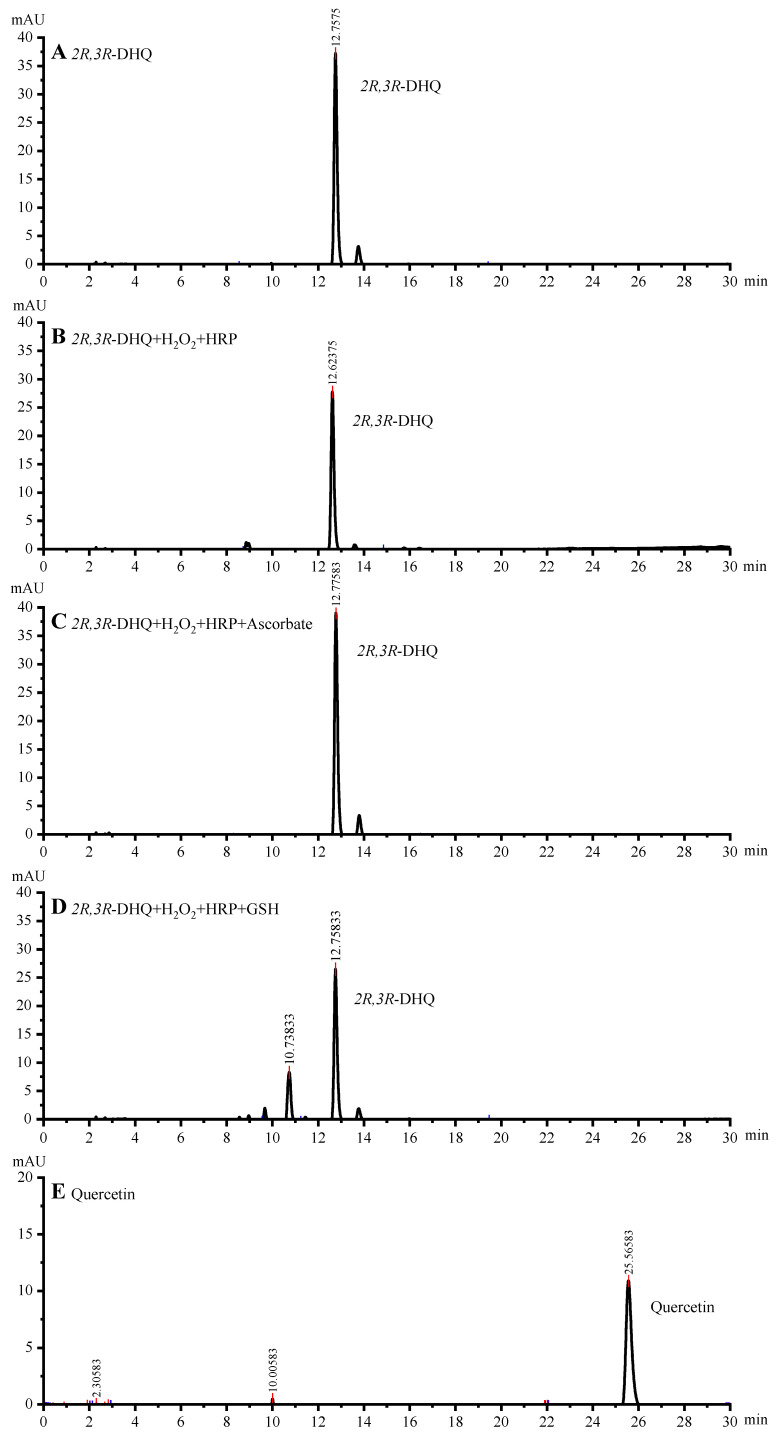
HPLC chromatograms of a 50 μM *2R*,*3R*-DHQ (**A**) and a 50 μM quercetin (**E**) calibration standard, and of the incubation mixture containing 50 μM *2R*,*3R*-DHQ, 3.2 nM HRP and 33 μM H_2_O_2_ (**B**), in presence of 50 μM ascorbate (**C**) or 100 μM GSH (**D**). The incubation mixtures were injected into the HPLC system 8.5 min after starting the reaction with HRP. The retention time of *2R*,*3R*-DHQ and quercetin were 12.7 min and 25.6 min, respectively. The consumption of *2R*,*3R*-DHQ in B and D were 14.4 ± 0.3 μM, and 12.1 ± 1.8 μM, respectively. A negligible consumption of *2R*,*3R*-DHQ was observed in (**C**) (0.04 ± 0.02 μM).

**Figure 4 ijms-24-14220-f004:**
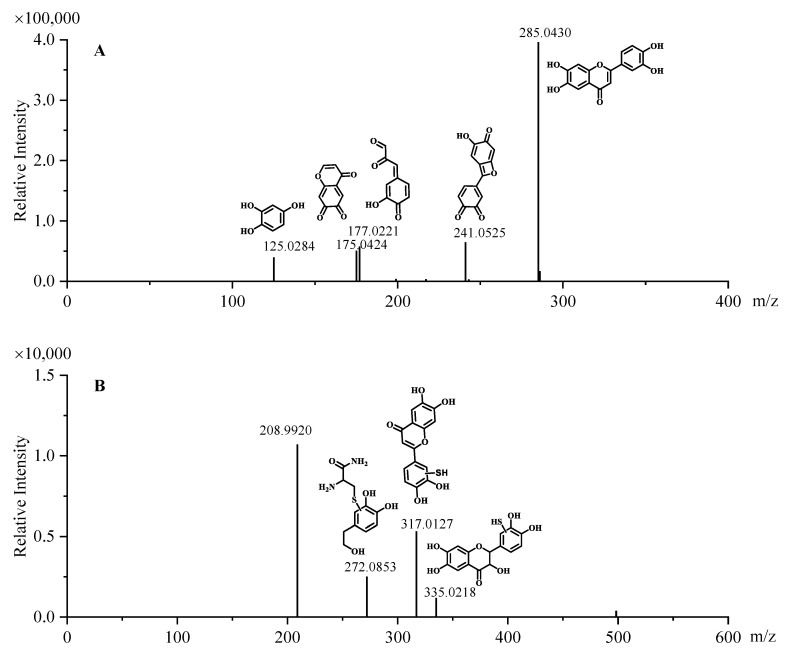
IT-TOF MS/MS spectra of *2R*,*3R*-DHQ (**A**) and the *2R*,*3R*-DHQ-GSH adduct (**B**) with characteristic fragment ions.

**Figure 5 ijms-24-14220-f005:**
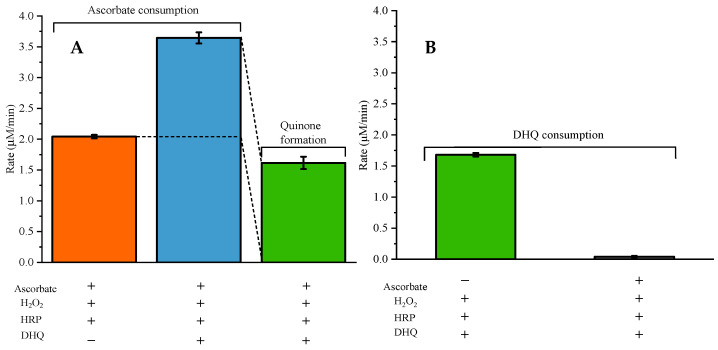
The rate of ascorbate consumption and *2R*,*3R*-DHQ quinone production (**A**) in the incubation mixture of 50 µM ascorbate, 33 µM H_2_O_2_, 3.2 nM HRP and when indicated, also containing 50 µM *2R*,*3R*-DHQ. The reaction was started with the addition of HRP. The consumption rate of ascorbate was calculated from the reduction in the absorption at 265 nm using a molar extinction coefficient of 12,840 M^−1^∙cm^−1^. The difference between the ascorbate consumption with and without *2R*,*3R*-DHQ was ascribed to reduction of the *2R*,*3R*-DHQ quinone generated by the HRP catalyzed oxidation of *2R*,*3R*-DHQ. This was used to estimate the rate of *2R*,*3R*-DHQ quinone formation. The rate of DHQ consumption (**B**) in the incubation mixture of 50 µM *2R*,*3R*-DHQ, 33 µM H_2_O_2_, 3.2 nM HRP with or without 50 µM ascorbate was calculated according to result of HPLC.

**Figure 6 ijms-24-14220-f006:**
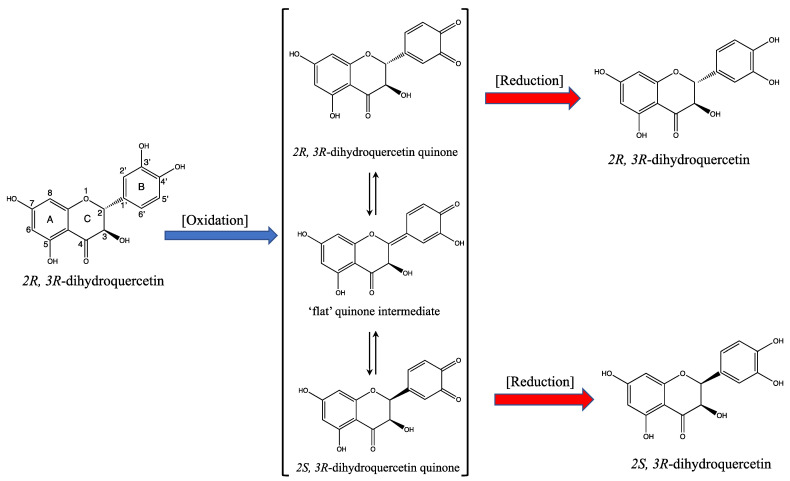
Reaction scheme of the potential epimerization of *2R*,*3R*-DHQ after oxidation by HRP/H_2_O_2_ and subsequent reduction of the DHQ quinone by ascorbate. *2R*,*3R*-DHQ is oxidized to form a quinone-like oxidation product. Three possible tautomers of the formed oxidation product are shown, e.g., *2R*,*3R*-DHQ quinone, quinone intermediate, and *2S*,*3R*-DHQ quinone. Reduction of *2R*,*3R*-DHQ quinone by ascorbate will lead to the formation of *2R*,*3R*-DHQ. Reduction of *2S*,*3R*-DHQ quinone by ascorbate will lead to the formation of *2S*,*3R*-DHQ. If the proposed tautomerization takes place by recycling, this will lead to the partial epimerization of *2R*,*3R*-DHQ into a mixture containing both *2R*,*3R*-DHQ and *2S*,*3R*-DHQ.

## Data Availability

The data presented in this study are available on request from the corresponding author.

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
