# Peer review of "Unraveling the Antioxidant Activity of 2R,3R-dihydroquercetin"

_ijms, 2023, doi:10.3390/ijms241814220_

Round 1

Reviewer 1 Report

The paper by Prof. Xu and colleagues reports a HPLC and MS-based study on the conversion of 2R,3R-dihydroquercetin to quercetin via a quinone intermediate. The formation of such quinone intermediate, generated upon reaction of 2R,3R-dihydroquercetin with hydrogen peroxide and HRP, was highlighted by its trapping with glutathione (GSH). The authors also demonstrated that in the presence of ascorbate the quinone intermediate – again formed from 2R,3R-dihydroquercetin, hydrogen peroxide and HRP – is reduced back to the parent dihydroquercetin derivative. The possible epimerisation of the quinone key intermediate was also considered and investigated through computer theoretical calculations.

Owing to the significant interest in the study of the reactivity and the reaction mechanism of antioxidants, I believe that the material reported in this manuscript can be, in principle, suitable for publication in Int. J. Mol. Sci. after revision. Indeed, there are some points that need to be addressed before publishing the manuscript and a major revision is required.

-          Attempts to characterise the quinon intermediate via NMR should be made in order to better investigate its structure. This could probably be done taking advantage of the higher stability of the GSH-quinone adduct.

-          Similarly, when considering the possible epimerisation of the quinone intermediate, NMR studies would enable to provide important data about the stereochemistry. Particularly, the adduct with SH should be investigated.

-          About the experimental investigation on the possible epimerisation of the quinone intermediate, the authors report that “If the epimerization of the quinone would take place and thus 2R,3R-DHQ quinone and 2S,3R-DHQ quinone are formed, the reduction of 2S,3R-DHQ quinone by ascorbate will yield 2S,3R-DHQ, and the reduction of 2R,3R-DHQ quinone by ascorbate will yield 2R,3R-DHQ. 2R,3R-DHQ and 2S,3R-DHQ can be identified separately using HPLC since they are diastereoisomers”. Did the authors run an HPLC control experiment using 2S,3R-DHQ as the standard sample or based this consideration on literature-reported HPLC studies on 2R,3R-DHQ and 2S,3R-DHQ? Two distereoisomers can have (accidentally) the same retention time under certain conditions. Thus, unless literature-reported studies on the separating efficacy of these conditions on both the considered diastereoisomers of DHQ, experiments with the 2S,3R-DHQ standard should be performed. Please, add and comment suitable literature-reported studies or provide the required experiments.

-          In some cases (i.e. page 8, line183) I would suggest to replace the term “racemization” with “epimerization” as the compounds considered by the authors are diastereoisomers and not enantiomers (only one out of two stereogenic center is ‘inverted’).

Some minor comments:

-          In the abstract I would suggest to replace the term “oxidator” with “oxidant”

-          All the stereochemistry indicator (R, S) should be in italic. Please, check throughout the whole manuscript.

-          Page 4: “Figure 1” should be labelled as “Figure 3”. Please, correct.

-          Page 9, Lines 208-211 and lines 213-216: please, delete the template text.

--

Reviewer 2 Report

The manuscript “Unraveling the antioxidant activity of 2R,3R-dihydroquercetin” presents a study evaluating antioxidant activity of 2R,3R-dihydroquercetin starting from Rogozhin hypothesis.  

The former experiment was reproduced, and further interesting data were provided by high performance liquid chromatography and tof mass spectrometry. The topic may be interesting for the readers and could be accepted for publication, however I have few suggestions:

-       There are too many sentences starting with “that…” resulting in redundant and prolix periods. English editing is required to provide a more fluent reading.

-       Line 24: add acronym (HRP) to horseradish peroxidase.

-       Line 46: change extend to extent.

-       Section 1. Introduction: Add a brief summary on antioxidant properties of quercetin

-       Line 89: change figure 1 to figure 3.

-       Lines 213-216: delete.

-       Line 218 Rogozhin et al.: Add Ref.

-       Lines 263-267: rewrite the sentence.

-       Please check and adjust the "Reference list" based on the regulations of reference list of journal. (Titles, doi, the name of journal and ... )

Moderate editing required

Reviewer 3 Report

This paper contains many flaws and does not deserve to be published in this journal.

The authors refer to the results shown in the Supplementary data, and this file is not attached at all, so it is not possible to fully verify all the results.

In part 3. Discussion, they did not remove the part from the template.

The conclusion is also very poorly written.

The paper needs to be reviewed in detail and some parts rewritten.

Moderate editing of English language required.

Round 2

Reviewer 3 Report

The manuscript is now eligible for publication.